# Cardiopulmonary Injury in the Syrian Hamster Model of COVID-19

**DOI:** 10.3390/v14071403

**Published:** 2022-06-27

**Authors:** Yi Xue, Dong Yang, Peter Vogel, Jennifer Stabenow, Lillian Zalduondo, Ying Kong, Yazhini Ravi, Chittoor B. Sai-Sudhakar, Jyothi Parvathareddy, Ernestine Hayes, Shannon Taylor, Elizabeth Fitzpatrick, Colleen B. Jonsson

**Affiliations:** 1Department of Microbiology, Immunology and Biochemistry, College of Medicine, University of Tennessee Health Science Center, Memphis, TN 38163, USA; yxue@uthsc.edu (Y.X.); ykong3@uthsc.edu (Y.K.); ehayes_1@yahoo.com (E.H.); efitzpat@uthsc.edu (E.F.); 2Regional Biocontainment Laboratory, University of Tennessee Health Science Center, Memphis, TN 38163, USA; dyang17@uthsc.edu (D.Y.); jstabeno@uthsc.edu (J.S.); lzalduon@uthsc.edu (L.Z.); jparvath@uthsc.edu (J.P.); stayl108@uthsc.edu (S.T.); 3Animal Resources Center and Veterinary Pathology Core, St. Jude Children’s Research Hospital, Memphis, TN 38105, USA; peter.vogel@stjude.org; 4Department of Surgery, University of Connecticut Health Center, Farmington, CT 06085, USA; ravi@uchc.edu (Y.R.); saisudhakar@uchc.edu (C.B.S.-S.); 5Institute for the Study of Host-Pathogen Systems, University of Tennessee Health Science Center, Memphis, TN 38163, USA; 6College of Pharmacy, University of Tennessee Health Science Center, Memphis, TN 38163, USA

**Keywords:** COVID-19, hamster, SARS-CoV-2, pulmonary vessels, pathology, heart, lung, blood

## Abstract

The Syrian hamster has proved useful in the evaluation of therapeutics and vaccines for severe acute respiratory syndrome-coronavirus-2 (SARS-CoV-2). To advance the model for preclinical studies, we conducted serial sacrifice of lungs, large pulmonary vessels, and hearts from male and female Syrian hamsters for days 1–4, and 8 post-infection (dpi) following infection with a high dose of SARS-CoV-2. Evaluation of microscopic lung histopathology scores suggests 4 and 8 dpi as prime indicators in the evaluation of moderate pathology with bronchial hyperplasia, alveolar involvement and bronchiolization being key assessments of lung disease and recovery, respectively. In addition, neutrophil levels, red blood cell count and hematocrit showed significant increases during early infection. We present histological evidence of severe damage to the pulmonary vasculature with extensive leukocyte transmigration and the loss of endothelial cells and tunica media. Our evidence of endothelial and inflammatory cell death in the pulmonary vessels suggests endothelialitis secondary to SARS-CoV-2 epithelial cell infection as a possible determinant of the pathological findings along with the host inflammatory response. Lastly, pathological examination of the heart revealed evidence for intracardiac platelet/fibrin aggregates in male and female hamsters on 8 dpi, which might be indicative of a hypercoagulative state in these animals.

## 1. Introduction

Animal models that recapitulate the multisystemic nature of infection and disease caused by severe acute respiratory syndrome-coronavirus-2 (SARS-CoV-2) in humans are critical for the interrogation of mechanisms underlying the host response, pathology and evaluation of approved or new investigational drugs that may alleviate disease [1]. The Syrian hamster has been widely accepted as a preclinical model for COVID-19 (coronavirus disease-2019) therapeutics and vaccines [2,3,4,5,6,7,8] based on the similarity of lung histopathology in human cases [9,10,11,12]. Prior to SARS-CoV-2, Syrian hamsters have been used as animal models for other human respiratory viruses [13], such as SARS-CoV [14], influenza virus [15] and adenovirus [16]. In silico comparison of the angiotensin-converting enzyme 2 (ACE2) sequence of humans with that of hamsters predicted that the SARS-CoV-2 S protein RBD would effectively bind hamster ACE2 [17], and provide a relevant animal model for COVID-19. While the COVID-19 hamster model has provided tremendous value in the assessment of vaccines and therapeutics, standardized models remain a gap. Published studies have used a wide range of timepoints for the assessment of virological and histopathological parameters, as well as viral strain, dose of infection, age and use of males or females [2,3,4,5,6,7,18].

Generally, intranasal infection of Syrian hamsters with SARS-CoV-2 USA-WA1/2020 strain results in mild clinical signs, such as rough hair coat and progressive body weight loss beginning 1–2 days post-infection (dpi) with recovery by 14 dpi [9,10,11]. While overt clinical signs appear mild, lungs show moderate disease with prominent or marked, multifocal lesions of pulmonary edema, inflammation, hemorrhage, alveolar damage, and cell death [10]. Micro-CT analysis reveals ground-glass opacities and evidence of gas in the cavity surrounding the lungs [11,19], similar to SARS-CoV infection in this model [14]. High-resolution micro-CT scans also demonstrate airway dilation and lung consolidations in infected hamsters [11,19]. High levels of the infectious virus have been reported in the lung and nasal turbinate on 2 and 4 dpi. Viral RNA and antigen have been detected in the small intestine with severe epithelial cell necrosis, deformed intestinal villi, and increased mononuclear cell infiltration [9]. Gruber et al., proposed a catalog of parameters to consider for standardizing reports of lung histopathology relevant to human disease. Of course, as underscored by Gruber et al. [12] the current knowledge regarding histopathology is from fatal human cases, and thus, translation to the COVID-19 hamster is difficult as lethality has not been reported. Important gaps in our knowledge of the COVID-19 hamster model remain in pathology studies of SARS-CoV-2 infected Syrian hamsters including limited descriptions of microscopic pathology of the lung and systematic evaluation of other organs, such as the heart and pulmonary vessels. Additionally, a systematic evaluation of the hematology of the COVID-19 hamster model has not been reported.

As reported by others, our evaluation of a low (10^3^ PFU) and high dose (2 × 10^5^ PFU) of SARS-CoV-2 USA-WA1/2020 strain in the Syrian hamster model resulted in greater lung pathology at the high dose Appendix A [10,13,20]. Of importance in this study was our preliminary finding that in the hamsters infected with the high dose there were vascular abnormalities, such as fibrin aggregates in pulmonary blood vessels, with necrosis and hemorrhage in surrounding areas (Appendix A). To investigate these findings further, and to advance the COVID-19 hamster model for preclinical studies, we used the higher dose and conducted a comprehensive, serial sacrifice study of male and female hamsters from 1–4 and 8 dpi with a focus on hematology, lung microscopic pathology, and pathologies of the pulmonary vessels, and heart. Of importance, we noted hematological abnormalities early in infection, increased RBC count and hematocrit. We present evidence of acute inflammatory cell infiltrates in the large pulmonary arteries and veins in male and female hamsters. Immunohistochemistry (IHC) of viral nucleoprotein antigen from the lung, pulmonary vessels and heart showed the bronchial epithelium as the major target for infection with minor staining in type 2 alveolar cells and no staining in the large pulmonary arteries and veins or heart. Pathological examination of the heart revealed evidence for intracardiac platelet/fibrin aggregates in 13–18% of male and female hamsters examined on 8 dpi, which might be indicative of a hypercoagulative state in these animals. In summary, we present several new findings that may serve as key endpoints for the assessment of disease in the blood, lung, and heart of male and female hamsters that will facilitate the evaluation of therapeutics and vaccines or mechanistic interrogations of disease pathways.

## 2. Materials and Methods

### 2.1. Cells and Virus

SARS-Related Coronavirus 2, Isolate USA-WA1/2020, NR-52281 was obtained through BEI Resources Centers for Disease Control and Prevention and, NIAID, NIH. The virus was amplified in Vero E6 TMPRSS cells using Minimum Essential Media with Earle’s salts (MEM) with 1% penicillin-streptomycin (P/S). Infection of cells was conducted at an MOI = 0.1 with 2% fetal bovine serum for 1 h and then MEM containing 1% P/S, and 10% FBS was added. Passage 3 viral seed stocks were sequenced and used in animal studies. Virus seed stock titers were measured using plaque assays with Vero E6 (ATCC CRL-1586). Vero E6 were maintained in Minimum Essential Medium (MEM) with Earle’s salts and L-glutamine. All reagents and cell culture reagents were purchased from Thermo Fisher Scientific unless specified.

### 2.2. General Animal Study Information

LVG Golden Syrian hamsters were supplied by Charles River. Hamsters were identified with ear tags and microchips (IPTT-300 by Bio Medic Data System). Hamsters were housed individually in caging under sealed, negative pressure in Allentown Biocontainment Unit (BCU-3) Rat racks. The microchips were implanted subcutaneously between the shoulder blades of the hamsters. All hamsters were anesthetized with ketamine (100 mg/kg) plus xylazine (10 mg/kg) by I.P. prior to intranasal infection. As specified, subsets of hamsters were anesthetized with ketamine (250 mg/kg) and xylazine (100 mg/kg) by intraperitoneal injection and euthanized via isoflurane anesthesia followed by a bilateral thoracotomy. Because the weights of hamsters vary, the volume of K/X injected was calculated based on the weight of the hamster. Clinical signs were monitored twice daily upon the manifestation of weight loss throughout the study. Otherwise, hamsters were monitored once per day upon recovery. The main clinical sign is weight loss; however, hamsters were evaluated for arched back and slightly rough coat (score = 1), weight loss and disinterest in food/water or changes/signs of respiratory distress (score = 2) and weight loss of greater than 10%, closed eyes or prostrate (score = 3 which requires euthanasia). If a hamster received a score of 2, a veterinarian was called. Hamsters showed minimal clinical signs and we have never observed scores above 1 for infection with SARS-CoV-2 WA1/2020. Hamsters were provided enrichment with one running wheel and one Igloo per cage both from bio-serve inc. Studies were conducted in accordance and approval by the Institutional Animal Care and Use Committee of the University of Tennessee Health Science Center (Protocol #20-0132).

### 2.3. Serial Sacrifice Study of Female and Male Hamsters

Within a B1 classified biosafety cabinet, twenty females at 9-weeks-old and twenty males at 7-weeks-old Golden Syrian hamsters were intranasally-infected with 2 × 10^5^ pfu of SARS-CoV-2 in a total volume of 100 μL. The ages differed based on the availability of males and females by the vendor but were within the range of 7–9 weeks of age used in our studies. Prior to infection, hamsters were anesthetized as described above. On 1, 2, 3, 4 or 8 dpi, blood, lung, and heart were harvested and snap frozen in liquid nitrogen or placed into formalin. Blood was collected for hematology as described below and analyzed the same day. The whole lung was collected, the left single lobe was taken for histopathology and the right five lobes were taken for measurement of viral titers by plaque assay as described below.

### 2.4. Hematology Analysis

Terminal percutaneous cardiac blood samples were obtained from each hamster after the induction of deep anesthesia with ketamine/xylazine. Blood was placed into Microtainer EDTA tubes, mixed according to the manufacturer’s instructions, and analyzed within 6 h of collection. A CBC (complete blood count) was performed using a DiaSys Xpedite™ HEM3 VET hematology analyzer (DiaSys Diagnostic Systems, Holzheim, Germany). Tri-level quality control material (Para12Extend Controls) was run daily prior to sample analysis. The Xpedite HEM3 analyzer provides direct counting for WBC, RBC and platelets using an impedance-based system, hemoglobin is photometrically measured, and all other parameters are calculated values: MCH (mean corpuscular hemoglobin), MCHC (mean corpuscular hemoglobin concentration), MCV (mean corpuscular volume), MPV (mean platelet volume) and HCT (hematocrit). A three-part white blood cell differential analysis is performed and reported as the absolute number and percentage of lymphocytes, MID (monocytes, eosinophils, and basophils) and granulocytes (neutrophils). All reagents were purchased from DiaSys, and analysis was performed according to the manufacturer’s established procedures.

### 2.5. Histology and Immunohistochemistry

Tissues were fixed in 10% neutral formalin for 48 h and embedded in paraffin; 5 µm serial sections were obtained and stained with H&E, NACE stain with 91C-1KT (Sigma-Aldrich, St. Louis, MO, USA), or by IHC for SARS-CoV-2 with rabbit anti-SARS-CoV-2 nucleoprotein (40143-R001, Sino Biological) or ACE-2 with rabbit anti-ACE-2 monoclonal Ab (NBP-67692, Novus Biological, CO, USA) Prosurfactant protein C (proSP-C) was detected similarly using rabbit anti-proSP-C from (Sigma-Aldrich AB3786). For immunostaining of SARS-CoV-2 nucleoprotein, slides were deparaffinized in xylene and rehydrated in decreasing concentrations of ethanol, and then treated with preheated Citrate-Based Antigen Unmasking Solution pH 6.0 (H-3300, Vector Laboratories) to reveal the nucleoprotein for 10 min at 97 °C as recommended by the manufacturer. Similarly, ACE2 was unmasked with Tris-based Antigen Unmasking Solution (H3301, Vector Laboratories, Newark, CA, USA) as recommended by the manufacturer. Unmasked tissue sections were rinsed with deionized water, endogenous peroxidase activity was quenched with 1% hydrogen peroxide. After rinsing 3× with phosphate buffer saline solution (PBS), slides were incubated overnight at 4 °C with primary rabbit antibody against nucleocapsid diluted at 1:1000 or ACE-2 diluted at 1:2000 (Cat # 40143-R001, Sino Biologicals, Beijing, China). Biotinylated secondary anti-rabbit IgG antibody was applied at 1:200 (Vector Laboratories) at room temperature for 1 h, rinsed and then incubated with Vectastain Elite using the ABC-HRP (Avidin-Biotin Complex-horseradish peroxidase) Kit from Vector Laboratories for 1 h (Vector Laboratories) at room temperature. Slides were counterstained with hematoxylin, dehydrated in ethanol and xylene and a coverslip mounted with Permont. Images were scanned with an Olympus SlideView VS200 using the Olympus OlyVIA v.3.2.1 imaging software.

### 2.6. Evaluation of Banked Heart Samples from Studies of Female and Male Hamsters

Four additional studies were conducted using the general parameters stated above and lung and heart tissues from sham inoculated (PBS) and SARS-CoV-2-infected were banked in paraffin blocks. In these studies, six- to nine-week-old male (*n* = 8) or female (*n* = 16) hamsters were purchased from Charles River. Hamsters were intranasally-infected with 2 × 10^5^ pfu of SARS-CoV-2 in a total volume of 100 μL. Tissues were collected and placed into 10% formalin as described for the serial sacrifice above. Using these banked specimens, we examined the heart tissue from sham or SARS-CoV-2 inoculated hamsters from 8 dpi. Evaluation of the banked tissues by histology and immunohistochemistry were as described above.

### 2.7. Microscopic Pathology Evaluation

Histopathological evaluation was completed on lungs obtained from hamsters at) 1, 2, 3, 4, and 8 dpi. Lungs were evaluated for lesions, including airway epithelium loss, exudates, bronchial epithelial hyperplasia, vascular inflammatory cell margination/endothelial hyperplasia, peribronchial inflammation and edema, septal thickening/inflammation, alveolar inflammation, alveolar edema/hemorrhage, extent of alveolar damage, pulmonary consolidation, and alveolar epithelial hyperplasia. Each of these types of pulmonary lesions was assigned a severity grade on a 1–5 scale as follows: 0 = no lesions; 1 = minimal, focal to multifocal, inconspicuous; 2 = mild, multifocal, prominent; 3 = moderate, multifocal, prominent; 4 = marked, multifocal or coalescing, lobar; 5 = severe, extensive, or diffuse, multilobar, with consolidation. Intermediate severity grades were also assigned as needed and these grades were then converted to weighted semi-quantitative scores as follows: 0 = 0; 1 = 1; 1.5 = 8; 2 = 15; 2.5 = 25; 3 = 40; 3.5 = 60; 4 = 80; 4.5 = 90; 5 = 100. These lesion scores used to calculate mean severity scores for each experimental group.

### 2.8. Plaque Assay

Plaque assays were used to measure the SARS-CoV-2 titer from seed stocks or homogenized tissues. Twelve-well plates were seeded with Vero E6 cells overnight. Ten series of 10-fold dilutions of virus samples were made in MEM with 2% FBS, and 200 μL of each dilution was added to each well in duplicate and incubated at 37 °C, 5% CO_2_ for 1 h. Following incubation, overlay media consisting of a 1:1 mix of 2% carboxymethylcellulose and 2XMEM supplemented with 10% FBS, 1% L-glutamine, 2% penicillin-streptomycin was added to each well. Plates were incubated at 37 °C, 5% CO_2_ for 72 h. Wells were fixed with 10% formalin, stained with 1% crystal violet, washed with Dulbecco’s phosphate buffered saline, and plaques were counted.

### 2.9. Data and Statistical Analyses

Body weight change, plaque forming units (viral load) and CBC (see hematology analysis above) were analyzed using GraphPad Prism 9 (GraphPad Software, Inc., La Jolla, CA, USA).

## 3. Results

### 3.1. Serial Sacrifice Study Design, Body Weight, Viral Load over Time

We conducted a serial sacrifice of males (*n* = 2) and females (*n* = 2) on 1, 2, 3, 4, and 8 dpi using the high dose, 2 × 10^5^ pfu, of the SARS-CoV-2 USA-WA1/2020 or PBS. Blood was taken for hematology. Lung and heart were taken for histopathology and immunohistochemistry of viral antigen as will be presented. Lungs were evaluated for microscopic pathology and scored.

At each time point, we recorded the corresponding body weight and viral load in the lung (Figure 1). Male and female hamsters dropped approximately 10% body weight by 5 dpi (Figure 1A). In both sexes, the viral load was essentially stable in the lungs from 1–4 dpi and generally decreased markedly on 8 dpi (Figure 1B). In one male, the virus titer remained high 8 dpi.

### 3.2. Hematology Showed a Significant Elevation of Neutrophils, RBCs, and Hemoglobin Early in SARS-CoV-2 Infection

The complete blood counts (CBC) revealed the most significant changes on 1 dpi (Figure 2). White blood cell measurements exhibited a significant increase in both the percentage (1 dpi, *p* = 0.0004; 2 dpi, *p* = 0.003) and absolute number (1 dpi, *p* = 0.03, Figure 2) of granulocytes. The WBC population was 65% neutrophils on 1 dpi followed by a decline to background levels by 4 dpi (Figure 2). Similarly, the percentage of lymphocytes in the blood was decreased at 1 dpi (*p* = 0.00008) and 2 dpi (*p* = 0.012) but returned to baseline by 3 dpi. Total RBC levels (*p* = 0.0004) and hematocrit (*p* = 0.0008) were significantly elevated from mock by 3 dpi. The hematocrit levels remained elevated from 1 to 8 dpi (Figure 2). Hemoglobin levels were significantly elevated from mock on 3 dpi (*p* = 0.003) and 4 dpi, (*p* = 0.03). Although not significant, platelet counts were decreased at 2 dpi but subsequently were similar to mock-infected through 8 dpi.

### 3.3. Evaluation of Microscopic Lung Pathology in Syrian Hamsters Show Peak of Inflammation and Lesions on Day 4 Post-Infection except for Bronchiolization

Microscopic lesions within the lungs of each hamster were examined and scored at each time point (Appendix A). Scores were combined to generate a heatmap (Figure 3). Bronchial/bronchiolar exudates and hyperplasia, endothelial hypertrophy/margination and peribronchiolar/perivascular inflammation were noted by 1 dpi with mild severity. Necrosis/apoptosis/sloughing of the bronchiolar epithelium was widespread on 2 dpi but minimal to absent at other time points. Bronchiolar epithelial hyperplasia was absent to mild on 1 dpi but increased markedly in severity and extent by 4 dpi, with some reduction in severity by 8 dpi. Minimal to mild alveolar inflammation was first detected on 2 dpi and continued to increase in extent and severity from 3 to 8 dpi. Mild to moderate pulmonary consolidation was first noted on 3 dpi and generally increased in extent and severity to 8 dpi. Endothelial hypertrophy associated with margination of inflammatory cells was widespread from 2 dpi to 4 dpi and was mostly resolved by 8 dpi. Similarly, the extent and severity of peribronchiolar/perivascular inflammation gradually increased from 1 dpi to 2–4 dpi but was declining by 8 dpi. Overall, the extent and severity of bronchiolar hyperplasia, alveolar and interstitial inflammation, and septal thickening were maximal on 4 dpi and were only slightly resolved by 8 dpi. Alveolar type 2 cell hyperplasia and/or bronchiolization of alveoli was absent until 8 dpi when this lesion was extensive in all four examined hamsters.

### 3.4. Viral Load, Neutrophil Presence and Lung Pathology in Syrian Hamsters Peaked at 4 dpi and Showed Resolution by Eight Days Post-Infection except for and Bronchiolization

Viral antigen (Figure 4, Figure 5, Figure 6, Figure 7 and Figure 8; see B, D, F, H, J) and immune cell infiltrates (Figure 4, Figure 5, Figure 6, Figure 7 and Figure 8; see B, D, F, H, J) were widespread in the bronchial epithelium and lower airways starting on 1 dpi and continuing through 2 dpi (Figure 4 and Figure 5; see E, C, G, I). On 2 dpi, a notable increase in debris and virus was noted in the lower airways (Figure 5C,E,G,I). By 3 dpi the numbers of ciliated cells in the large airways were reduced and disorganized (Figure 6F). We noted hyperplasia in the bronchial epithelia on 3 dpi. Consolidation was apparent around blood vessels (Figure 6I) with colocalization of the virus in consolidated areas of alveolar epithelial cells (Figure 6J). Hyperplasia was observed at 4 dpi with bronchiolization as well as the proliferation of the bronchial epithelial and pneumocytes (Figure 6C,I) and vasculitis (Figure 7J). Neutrophils were observed in airways over the first 4 dpi (Figure 4, Figure 5, Figure 6 and Figure 7; see E). By 8 dpi, virus antigen was absent (Figure 8B,D,F,H) and there was decreased immune cell infiltration and neutrophils in the bronchial epithelium (Figure 8E, J). The bronchial epithelium continued to show hyperplasia (Figure 8C,D). Eosinophils were also present in the alveolar spaces (Figure 8F).

### 3.5. Pathology of Pulmonary Arteries and Veins in Syrian Hamsters Were Characterized by Extensive Leukocyte Transmigration

The large pulmonary arteries and veins were evaluated on 1, 2, 3, 4 and 8 dpi by H&E (Figure 9 and Appendix A). By 1 dpi, leukocytes were attached to the endothelium of pulmonary veins in all cases and the artery of one female hamster. An increased number of leukocytes penetrated through endothelial cells and the tunica intima by 2 dpi. By 3 dpi, neutrophils were abundant in the tunica media of the pulmonary arteries and veins. Inflammatory cells continued to infiltrate the endothelial cell layer of the arteries on 4 dpi. In the pulmonary veins on 4 dpi, we noted continued infiltration of inflammatory cells and active mitosis in endothelial cells. We also noted clusters of cells with single nuclei in the tunica media (smooth muscle cells). By 8 dpi, there was decreased inflammatory cell infiltration in blood vessels and a higher density of endothelial cells in the tunica intima.

### 3.6. Intracardiac Platelet and Fibrin Aggregates in the Heart

In addition to the samples herein, we examined banked paraffin-embedded heart tissues from four studies using 6–9-week-old male (*n* = 8 SARS-CoV-2, *n* = 2 PBS-inoculated) or female (*n* = 18 SARS-CoV-2, *n* = 6 PBS-infected) hamsters. The body weight and viral load were measured for the four studies (Appendix A). Across these studies, sagittal sections of the heart were evaluated by H&E and probed by IHC for the nucleocapsid antigen. No viral antigen was detected in over 80 different sections in several studies (Figure 10C). Mild degeneration of the heart muscle was noted as soon as 1 dpi with the presence of vacuoles within cardiomyocytes in the right ventricle (Figure 10A) and abnormal appearance of the coronary artery muscle layer (Figure 10B). However, no other findings were noted with the SARS-CoV-2 WA20 strain in the examination of tissues across these studies until 8 dpi. In one study of female hamsters, two of ten specimens of heart on 8 dpi showed platelet and fibrin aggregates in the right ventricle and atrium. In one animal, this is shown for the right ventricle (Figure 10E) and both WBCs and RBCs were noted within the fibrin close to the tricuspid valve (higher magnification) (Figure 10F). In the second hamster, we show the right atrium, but similar findings were noted in the right ventricle (Figure 10G). Lastly, in a third, female hamster (8 dpi), we noted an abnormality in the endothelial cells with hyperplasia and WBCs near the damaged area (mononuclear cells and eosinophils) at the mitral valve (Figure 10H). Examination of heart specimens from female animals on 8 dpi from three additional studies suggested one additional animal with platelet and fibrin aggregates out of a total of eight. Lastly, in two studies of male hamsters infected with SARS-CoV-2, one of eight hearts was noted to have platelet and fibrin aggregates. In summary, evidence for intracardiac platelet/fibrin aggregates was noted in male (13%) and female (17%) hamsters on 8 dpi, which might be indicative of a hypercoagulative state in these animals.

## 4. Discussion

Given the global impact of COVID-19 pneumonia, a preclinical animal model that recapitulates the disease is critical for the evaluation of therapeutics and vaccines [1,21,22,23]. The results of this study of SARS-CoV-2 infection in male and female hamsters provide several parameters that can enable the evaluation of therapeutics and vaccines or mechanistic interrogations of disease pathways. In the context of a serial sacrifice study, we present evidence of hematological abnormalities, such as increased RBC count and hematocrit. Future studies will assess if this was driven due to dehydration and decreased oxygen levels. Neutrophil levels were pronounced in whole blood on 1 and 2 dpi (Figure 2), around blood vessels on 2 dpi (Figure 5I), penetrating the tunica media on 3 dpi (Figure 9) in male and female animals and were present in lung debris, the epithelium and submucosal layer by 1 dpi (Figure 4E) and 2–4 dpi (Figure 5, Figure 6 and Figure 7E). By 8 dpi, neutrophils were no longer present in the mucosa or submucosa (Figure 8E) but were present in the interstitial tissues of the lung along with eosinophils (Figure 8F). Monocytes and eosinophils were noted at 4 dpi around blood vessels with proliferating endothelial cells (Figure 7J).

We observed blood vessel injury with the loss of endothelial cells and tunica media, and diffuse damage in surrounding alveoli, pulmonary hemorrhage, and destruction of alveolar septa. This corroborates with the lung pathophysiology observed in human COVID-19 cases with significant multi-focal lesions indicative of moderate disease [9,10,11]. Endothelial injury of the pulmonary vessels is a principal determinant of microvascular dysfunction [24]. It results in shifting the vascular equilibrium towards vasoconstriction with subsequent organ ischemia, and inflammation, with associated tissue edema [24,25]. Our findings show the accumulation of inflammatory cells, with evidence of endothelial and inflammatory cell death, and suggest endothelialitis secondary to SARS-CoV-2 epithelial cell infection as a possible determinant of the pathological findings along with the host inflammatory response [24,25]. Immunothrombosis (i.e., upregulation of the immune and coagulation systems to block pathogens) driven by endothelialitis may be a major driving factor in lung injury observed in SARS-CoV-2 infection [24,25]. In the review by Bonaventura et al. [24], they propose that clinical manifestations of COVID-19 are driven by exaggerated immunothrombosis within the lung microvessels. In addition to regulating vascular tone, endothelial cells are critical for regulating thrombosis and maintaining an anti-thrombotic state. Currently, there are several proposed mechanisms of lung vascular disease and thrombosis formation during SARS-CoV-2 infection that include inflammation and cellular injury, NETosis (neutrophil extracellular traps (NETs) formation), complement activation, and injury mediated by platelet activation [26,27]. Our findings show an accumulation of inflammatory cells, with evidence of endothelial cell death. We did not observe direct infection of endothelial cells in areas where there was an accumulation of inflammatory cells and injury. Endothelialitis has been reported in one other study at 5 dpi in the COVID-19 hamster model with no detectable viral RNA [12]. Exposure of endothelial cells to lytic cell debris and proinflammatory cytokines can upregulate adhesion molecules to allow for the recruitment of leukocytes and platelets [26]. A lack of tropism for endothelial cells and the requirement of exposure to serum proteins may limit our ability to elucidate mechanisms of disease using in vitro assays.

Initial low platelet levels observed in our study on 2 dpi, may be due to platelet apoptosis [28]. Platelet count in human cases of COVID-19 may be dependent on the severity of infection [28,29]. In the serial sacrifice study, mild degeneration of the heart muscle was observed by 1 dpi, however, no other pathologies were detected. We expanded our analyses of the heart to include additional PBS or SARS-CoV-2-inoculated samples that we have banked as paraffin blocks from the start of the pandemic. In these four studies, male and female hamsters were intranasally infected with the 2 × 10^5^ PFU of SARS-CoV-2 in the 6–9-week-old age range. Intracardiac platelet and fibrin aggregates were detected in 13–17% of the male and female hamsters (Figure 10). In these specimens, we observed the platelet and fibrin in the right atrium and right ventricles. The PBS-inoculated hamsters from the studies reported herein, or from these studies, had no platelet and fibrin or any pathology. Viral antigen was not detected in over 80 sagittal sections from these studies. These findings suggest a hypercoagulative state in these animals. Moreover, our findings provide a premise to expand our examination of inflammatory markers, and coagulation pathways, to define the mechanisms driving the progression of heart failure to the chronic stages and the potential involvement of the pulmonary right ventricular axis.

RBC alterations and thrombosis in COVID-19 disease need to be further investigated in animal models. While the incidence of cardiac abnormalities was low in the 6–9-week-old COVID-19 hamster model, it is possible that this may increase in older hamsters or hamsters with comorbidities, such as diabetes. As our studies used 6- to 9-week-old hamsters, our future studies will evaluate incidence in older age groups with the approaches and markers identified herein [18]. Further development of COVID-19 animal models is important for the elucidation of mechanisms and to provide a strong premise for the development of therapies and interventions targeting cellular and molecular mechanisms involved in the development of cardiac dysfunction which is key to the progression and severity of the COVID-19 disease process [30,31]. Our findings in the hamster model (with mild/moderate infection) are reflective of the pathology observed in SARS-CoV-2 infection of human subjects in the acute stages. However, patients with mild/moderate infection are not severely ill, thereby limiting access to the availability of specimens for histopathological analysis. We observed in our model early evidence of cardiopulmonary failure, and early thrombus formation. Further work to elucidate the beneficial effects of the early institution of anti-platelet, anti-coagulant, fibrinolytic and/or thrombolytic therapy to ameliorate the prothrombotic and procoagulant state would be of significant benefit and lead to further evaluation in afflicted human subjects and have significant translational potential. Finally, understanding the complexity and progression of cardiac involvement using additional approaches, such as echocardiography of the hamster-COVID-19 model, will advance translation [30,31].

## Figures and Tables

**Figure 1 viruses-14-01403-f001:**
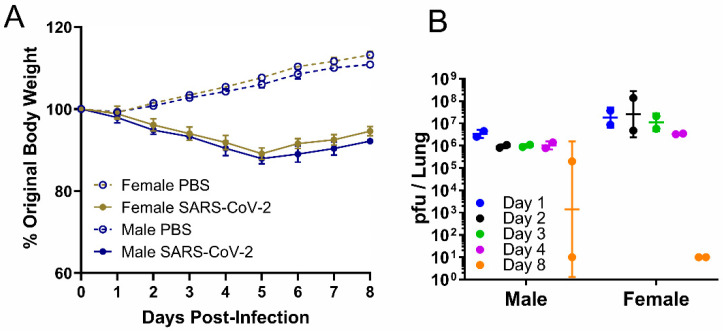
Weight and viral load in male or female hamsters intranasally infected with PBS or SARS-CoV-2 at a high dose, 2 × 10^5^ pfu. In the serial sacrifice study of SARS-CoV-2-infected hamsters, (**A**) body weight change (data are presented as mean ± SEM) was measured daily from 0 to 8 dpi, and (**B**) viral load was measured by plaque assay in lungs of male and female hamsters at five time points (*n* = 2). Data are presented as geometric mean ± geometric SD.

**Figure 2 viruses-14-01403-f002:**
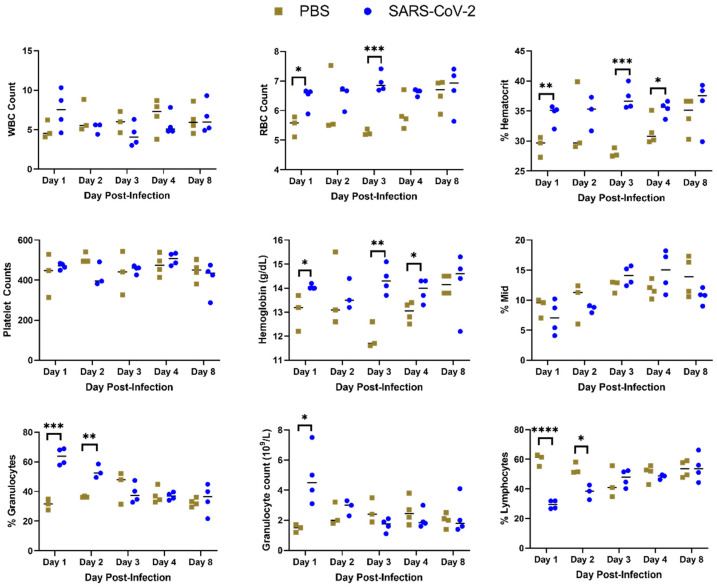
Complete blood count from hamsters infected with SARS-CoV-2 or sham inoculated (PBS). Blood was drawn from hamsters on 1, 2, 3, 4 or 8 dpi and a complete blood count was performed. (*: *p* < 0.05, **: *p* < 0.01; ***: *p* < 0.001, ****: *p* < 0.0001.)

**Figure 3 viruses-14-01403-f003:**
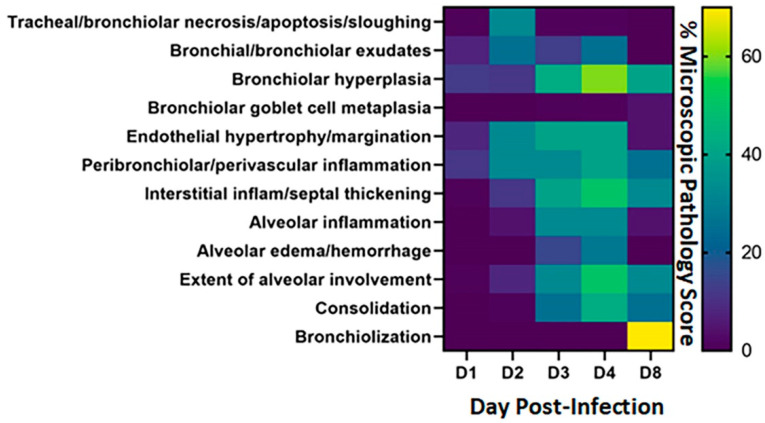
Heatmap of microscopic pathology of Syrian hamster lungs with SARS-CoV-2. The severity scores for each category of microscopic pathology are illustrated using a heatmap. For each of the days examined, each box represents the average score from four hamsters for scores for these categories: tracheal/bronchiolar necrosis/apoptosis/sloughing, bronchial/bronchiolar exudates, bronchiolar hyperplasia, bronchiolar goblet cell metaplasia, endothelial hypertrophy/margination, peribronchiolar/perivascular inflammation, interstitial inflammation-septal thickening, alveolar inflammation, alveolar edema/hemorrhage, extent of alveolar involvement, consolidation and bronchiolization. Semi-quantitative scores used 0–1 = within normal limits, 2–14 = Minimal: Rare or inconspicuous lesions; 15–39 = Mild: Multifocal or small, focal, or widely separated, but conspicuous lesions; 40–79 = Moderate: Multifocal, prominent lesions; 80–99 = Marked: Extensive to coalescing lesions or areas of inflammation with some loss of structure and 100 = Severe: Diffuse lesion with effacement of normal structure. The scale to the right of the heat map reflects the scoring ranges from 0 to 70%.

**Figure 4 viruses-14-01403-f004:**
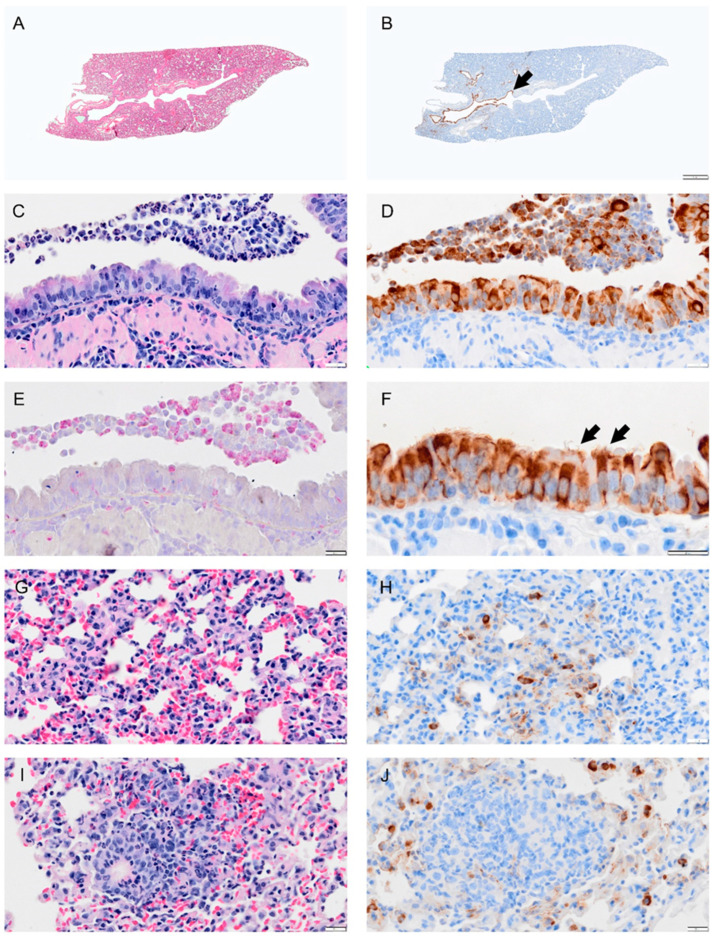
Selected pathology and immunohistochemistry of lungs from Syrian hamsters at 1 dpi with SARS-CoV-2. (**A**) Lower magnification of a H&E-stained section showed no marked change at 1 dpi. (**B**) Immunohistochemistry (IHC) of SARS-CoV-2 in lung section shows distribution of the virus in the bronchial epithelium of primary and secondary (segmental) bronchi (brown color arrowhead) and in some scattered areas of alveolar tissues. (**C**) H&E-stained section of bronchus showing inflammatory cell infiltration in bronchial wall (mucosa and submucosa) and cell debris (neutrophils, macrophages, necrotic epithelial cells) in lumen. (**D**) Adjacent serial sections of panel (**C**) show abundant infected epithelial cells in mucosa and debris. (**E**) Adjacent serial section with NACE stain shows neutrophils in debris, epithelium, and submucosal layer. (**F**) Higher magnification of virus IHC shows disorganized bronchial epithelial cells with disorganized cilia, see arrow. (**G**) H&E-stained section of alveolar tissue showing intravascular neutrophils and congestion. (**H**) Adjacent section of panel (**G**) stained for virus by IHC shows virus antigen concentrated in type 2 alveolar cells, with small amounts lining alveolar surfaces. (**I**) Section showing inflammatory cells in areas of alveolar collapse, representing early consolidation. (**J**) Viral antigen concentrated within type 2 pneumocytes surrounding consolidation area. Bar: (**A**,**B**) = 1 mm; (**C**–**J**) = 20 µm.

**Figure 5 viruses-14-01403-f005:**
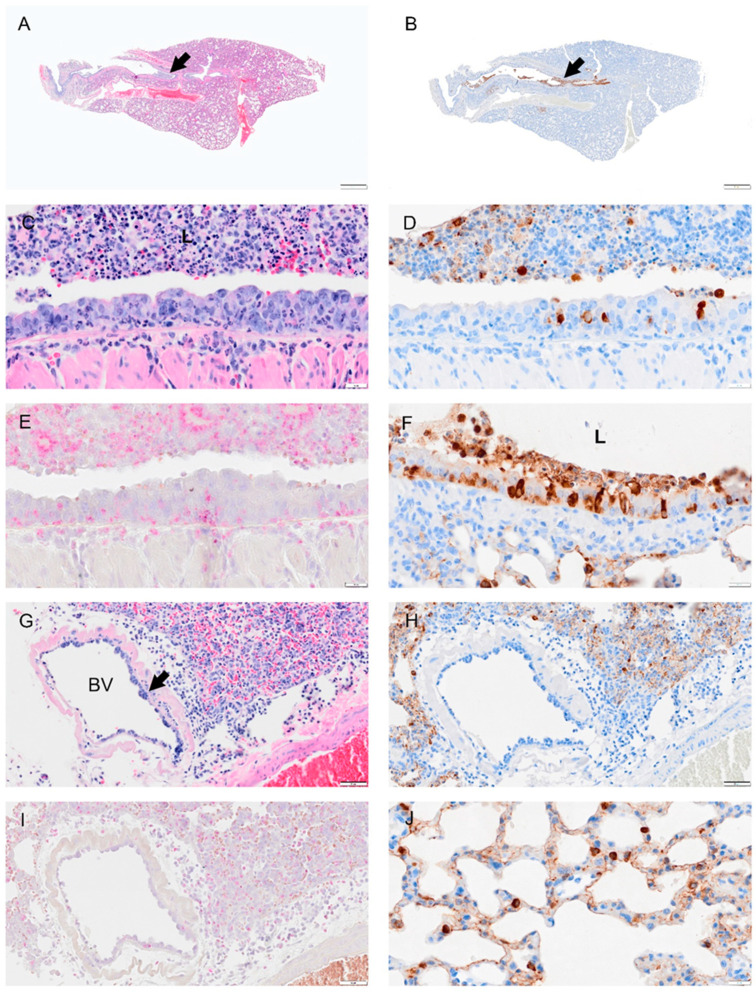
Selected pathology and immunohistochemistry of lungs from Syrian hamsters at 2 dpi with SARS-CoV-2. (**A**) A low magnification image shows widespread septal thickening and debris in the primary bronchial lumen (arrow). (**B**) Image of serial section adjacent to (**A**) showing viral antigen bronchial epithelium and debris. (**C**) Section showing the bronchus with abundant cell debris (neutrophils, monocytes, necrotic epithelial cells, pyknosis and karyorrhexis, red blood cells) in the bronchial lumen (L). Image also shows damaged epithelial cells (loss of cilia, disruption of cell layer and inflammatory cells in mucosal and submucosal layer. (**D**) Serial section adjacent to (**C**) showing scattered viral antigen positive cells in the epithelial layer and in lumenal debris. (**E**) NACE staining of bronchus which shows many infiltrating neutrophils. (**F**) Viral antigen in the bronchus, debris of lumen (L) and type 2 alveolar cells. (**G**) A blood vessel (BV) showed thickened hypercellular endothelium(arrow) and inflammatory cell infiltrates in the vascular tunica media, interstitium, and alveoli. (**H**) Serial section adjacent to (**G**) shows viral antigen in the alveolar cells but not in the blood vessel. (**I**) NACE staining of the blood vessel and surrounding tissues showing neutrophil infiltration. (**J**) Section shows viral antigen in type 2 alveolar cells and thickened septa. Bar: (**A**,**B**) = 1 mm; (**C**–**F**) = 20 µm; (**G**–**I**) = 50 µm and (**J**) = 20 µm.

**Figure 6 viruses-14-01403-f006:**
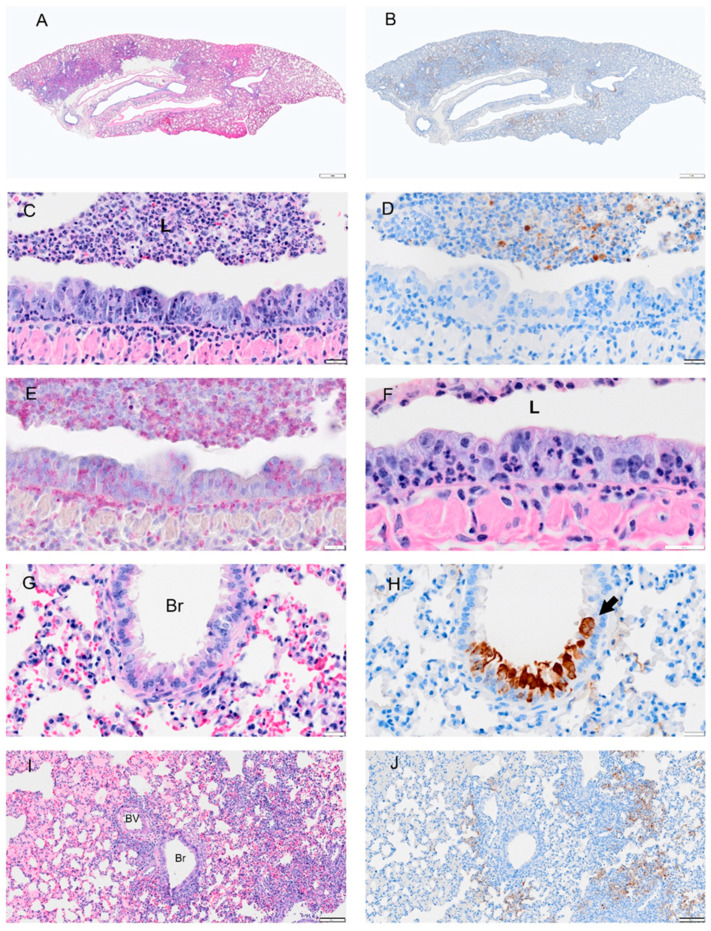
Selected pathology and immunohistochemistry of lungs from Syrian hamsters at 3 dpi with SARS-CoV-2. (**A**) Low magnification of a section with multifocal areas consolidation and hemorrhage. (**B**) An adjacent serial section of (**A**) shows viral antigen is concentrated at the margin of consolidated areas. (**C**) Higher magnification of bronchus shows debris with necrotic neutrophils, epithelial cells, and macrophages in bronchial lumen (L), abundant neutrophils within epithelial layer and submucosa, hyperplastic epithelial cells, and decreased cilia. (**D**) Adjacent serial section of (**C**) showing absence of viral antigen in the bronchial epithelium and but some remaining in luminal debris. (**E**) An adjacent serial section of (**C**,**D**) shows abundant neutrophils in the luminal debris, and within the epithelial and submucosal layers (visualized using NACE stain). (**F**) High magnification of the bronchial epithelium shows damaged mucosal epithelium with absent or very short cilia; neutrophils were the primary type of inflammatory cells. (**G**) H&E-stained section shows degenerated epithelial cells in small airways (Br). (**H**) An adjacent serial section to (**G**) shows viral antigen in damaged epithelial cells (arrow). (**I**) H&E-stained section shows hyperplasia of bronchiolar epithelium (Br), intra-alveolar inflammatory cell infiltrates (right side of image), and serous exudates filling alveolar spaces (left side of image). (**J**) Adjacent serial section of (**I**) shows viral antigen tends to be concentrated at the periphery of consolidated areas. Scale bars: (**A**,**B**) = 1 mm; (**C**–**H**) = 20 µm; (**I**,**J**) = 100 µm.

**Figure 7 viruses-14-01403-f007:**
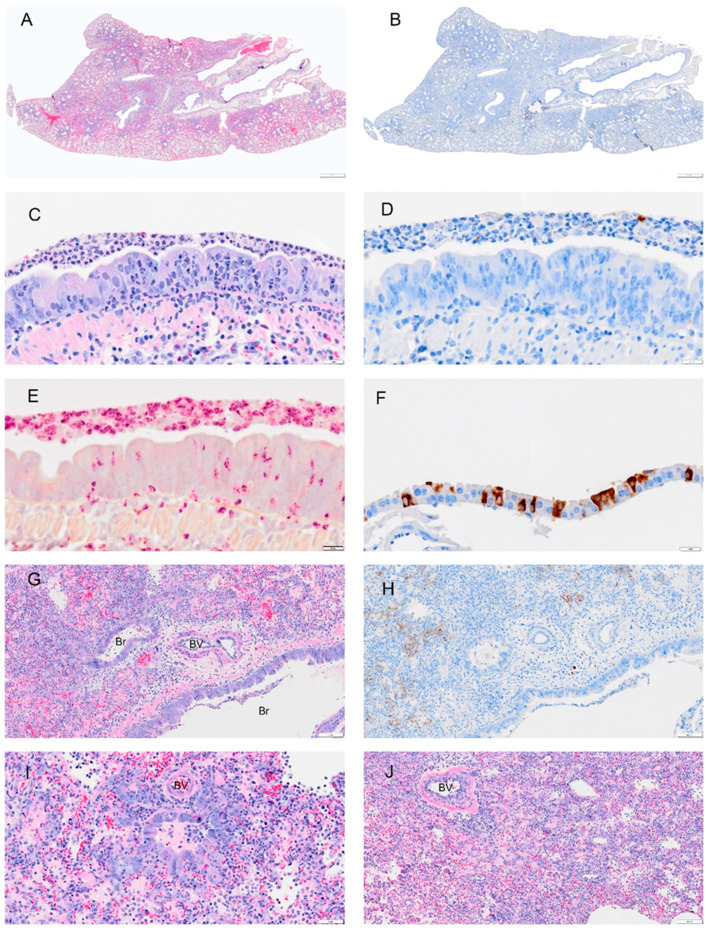
Selected pathology and immunohistochemistry of lungs from Syrian hamsters at day 4 post-infection with SARS-CoV-2. (**A**) Section of the lung shows more extensive consolidation. (**B**) Image showing overall reduction in viral antigen (not present in bronchi). (**C**) Higher magnification of bronchus shows decreased debris in bronchial lumen, abundant neutrophil infiltration in hyperplastic (and disorganized) epithelium and submucosa. (**D**) A serial section adjacent to (**C**) shows minimal viral antigen in epithelial layer and debris. (**E**) An adjacent serial section to (**C**,**D**), shows neutrophils in debris, epithelial and sub mucosa layer following NACE staining. (**F**) Virus antigen is still present in some bronchiolar epithelial cells. (**G**) H&E-stained section which shows hyperplastic bronchial epithelium with neutrophil infiltration, debris in lumen, necrosis (Br) and inflammatory cells infiltrates in lung parenchyma, a blood vessel with thickened endothelium (BV) and perivascular edema. (**H**) An adjacent serial section to (**G**) shows small pockets of viral antigen in pulmonary parenchyma (consolidation) but not in bronchial epithelium. (**I**) H&E-stained lung tissue shows perivascular (BV) and peri-bronchial inflammatory cell infiltration, alveolar exudates, and alveolar epithelial hyperplasia with mitosis. (**J**) A BV with proliferated endothelial cells Monocytes and eosinophils, and fibroblasts (undergoing mitosis) are distributed around blood vessels. Necrotic lung tissues show damaged alveolar walls and inflammatory cell infiltrates in consolidation area. Bar: (**A**,**B**) = 1 mm; (**C**–**F**) = 20 µm; (**G**,**H**,**J**) = 100 µm; (**I**) = 50 µm.

**Figure 8 viruses-14-01403-f008:**
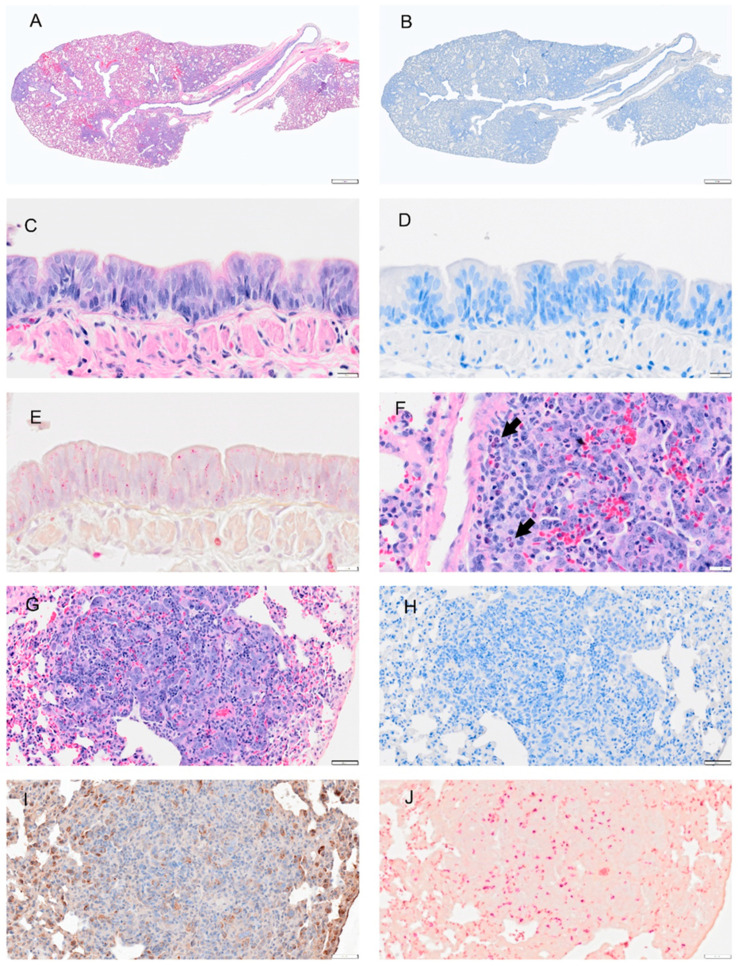
Selected pathology and immunohistochemistry of lungs from Syrian hamsters at 8 dpi with SARS-CoV-2. (**A**) H&E section shows well demarcated areas of consolidation and multifocal hemorrhage. (**B**) A representative section of the lung was subjected to IHC for the discovery of viral antigens. No staining for viral antigen was observed. (**C**) A higher magnification image of bronchial epithelial cells shows hyperplasia and few inflammatory cells (**D**). No viral staining was detected in adjacent serial section to (**C**). (**E**) A representative image stained with NACE stain shows no neutrophils in mucosa or submucosa. (**F**) H&E imaging shows increased eosinophils and neutrophils in the interstitial tissues (arrow) of the lung. Alveolar epithelial are hypertrophic and hyperplastic. (**G**) Representative section of the lung showing alveolar bronchiolization and inflammatory cell infiltration. (**H**) An adjacent section of (**G**) probed by IHC for viral antigen. No viral antigen was detected. (**I**) An adjacent serial section to (G/H) probed for prosurfactant protein C. IHC shows positive signals in cytoplasm of proliferating type 2 alveolar cells. (**J**) An adjacent serial section to (G/H) stained with NACE stain and shows neutrophils in these foci. Scalebar: (**A**,**B**) = 1 mm; (**C**–**F**) = 20 µm; (**G**–**J**) = 50 µm.

**Figure 9 viruses-14-01403-f009:**
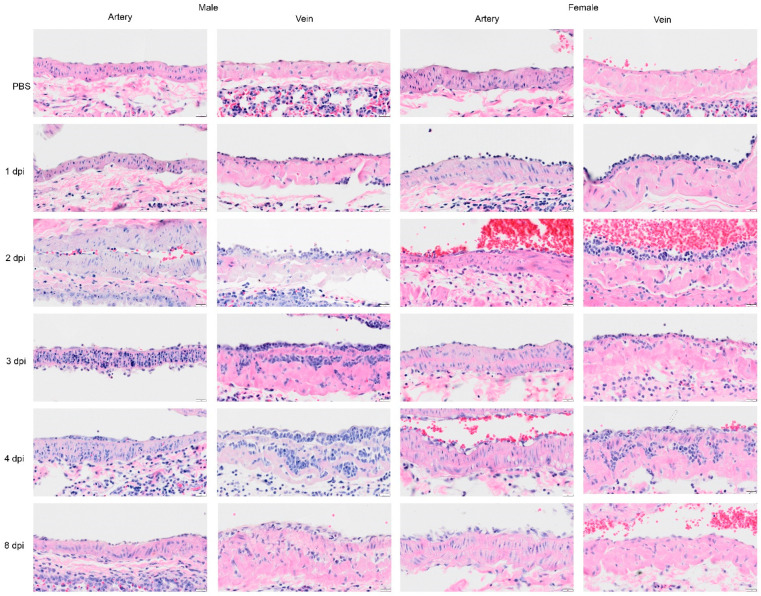
Selected H&E images of the large pulmonary blood vessels from Syrian hamster following infection with SARS-CoV-2 on 1, 2, 3, 4, and 8 dpi. In the sham inoculated group (PBS), the main pulmonary artery and veins display a normal character in each layer of the blood vessel. In subsequent rows, representative images of pulmonary artery and veins are presented for 1, 2, 3, 4, and 8 dpi. The pulmonary artery at 1 dpi is normal, although in one female hamster, a few leucocytes were attached to the tunica intima. At 1 dpi, the image of the pulmonary vein shows leucocytes attached to the wall of pulmonary veins. At 2 dpi, blood vessels show an increased level of inflammatory cells or leucocytes penetrating through endothelial cells, and tunica intima, and appeared to be greater in the vein. The image on 3 dpi shows neutrophils that have penetrated through tunica media of pulmonary artery (male) or less severe (female). In the image of the pulmonary veins, the inflammatory cells have clotted in the tunica media. The image on 4 dpi shows inflammatory cells passing through an intact endothelial cell layer of the artery (male) although some vacuoles are present in endothelial cells and in muscle cells (female). The image of the vein shows the infiltration and proliferation of mononuclear cells. Mitosis was also noted in the endothelial cells (arrow in female vein implies proliferation of endothelial cells). We also noted groups of increased cells with single nuclei in the tunica media (smooth muscle cells). By 8 dpi, there was a decrease in the level of inflammatory cells infiltrating the blood vessels (recovered), and a higher density of endothelial cells in tunica intima. Scalebar = 20 μm.

**Figure 10 viruses-14-01403-f010:**
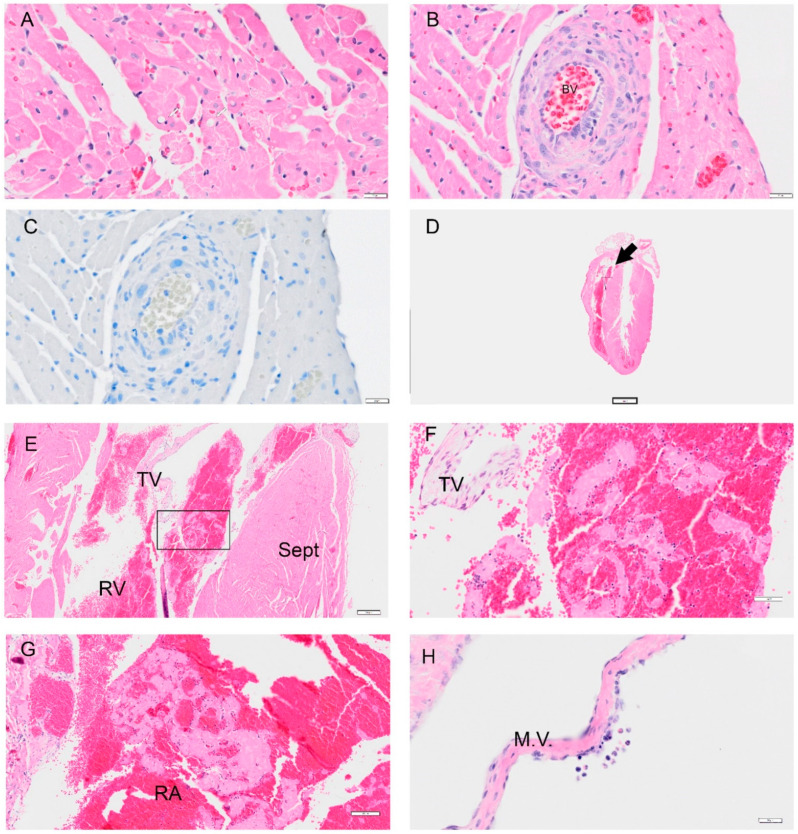
Selected pathology and immunohistochemistry of the heart from the Syrian hamster following infection with SARS-CoV-2. Representative images of pathology are presented. (**A**) An image at 1 dpi from the right ventricle showed mild degeneration of cardiomyocytes. (**B**) An image from 1 dpi shows an abnormal muscle layer in the coronary artery. (**C**) A representative image of the heart probed for viral antigen by IHC Viral N protein was not detected in over 80 slides. (**D**) A low magnification image of the heart at 8 dpi in a female hamster with an arrow showing a thrombus (0.4×). (**E**) An image of a right ventricle with a platelet and fibrin aggregate (4×). (**F**) Higher magnification of (**E**) which in the right ventricle with white blood cells (WBC) trapped in fibrin and red blood cells (RBC); close to the tricuspid valve (TV). (**G**) An image showing a thrombus in the right atrium (RA) with similar findings in the right ventricle in (**F**). (**H**) Image showing endothelial cell abnormality with hyperplasia of the endothelial cells. Image shows WBC (mononuclear cells and eosinophils) around a damaged area on the mitral valve (M.V.) (female 8 dpi). Abbreviations: RV: right ventricle; Sept: septum; TV: tricuspid valve. Scalebar: (**A**–**C**,**H**) = 20 μm; (**D**): = 2 mm, (**E**) = 200 μm; (**F**) = 50 μm, (**G**) = 100 μm.

## Data Availability

The data presented in this study are available in this article and Appendix A.

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
