# Peer review of "Cardiopulmonary Injury in the Syrian Hamster Model of COVID-19"

_viruses, 2022, doi:10.3390/v14071403_

Round 1

Reviewer 1 Report

Introduction part: The authors might correlates the progression of the disease in the animal model and the events described in humans; it should be done in a paragraph, now is distribute in different lines. 

Please explain deeper "Immunothrombosis driven by endothelialitis maybe a major driving factor in lung injury observed in this viral infection" It is stated some times in the MS without any additional explanation.

Author Response

Response: Thank you for this critique. We have modified this in the discussion where we discuss this possibility as follows: Immunothrombosis (i.e. upregulation of the immune and coagulation systems to block pathogens) driven by endothelialitis maybe a major driving factor in lung injury observed in SARS-CoV-2 infection [24,25]. In the review by Bonaventura et al. [24], they propose that clinical manifestations of COVID-19 are driven by an exaggerated immunothrombosis within the lung microvessels.

Reviewer 2 Report

The studies presented within this manuscript are important for furthering SARS-CoV-2 research.  However, the manuscript suffers from a lack of careful editing.  Further, the authors include some speculative statements that require clarification.  In addition, the inclusion of data from other studies is very concerning.  Much more detail needs to be provided regarding the methodology.  The following is a list of representative issues that must be addressed before a full evaluation of the manuscript on the basis of its scientific merit, can be made--note that this is likely not a complete list and that additional issues may be present and require attention.  The authors are encouraged to have one or two scientists and/or scientific technical writers not directly connected with the work, perform a thorough evaluation.

Lines 39-40—Insert comma and delete, “and” so that portion of sentence reads:  “…host response, pathology and evaluation…”

Line 47—either remove the word, “would” or change “binds” from plural to singular so portion of sentence reads:  “…RBD effectively binds….” or “…RBD would effectively bind….”

Lines 48-52—Break into two sentences.  “While the COVID-19 hamster model has provided tremendous value in the assessment of vaccines and therapeutics, standardized models remains a gap.”  [Note also that the word “remains” should be singular not plural.]  Then eliminate, “as” and start next sentence:  “Published studies have….”

Line 55—Wording is poor and need to correctly write out and then define dpi.  Revise portion of sentence to:  “…beginning 1-2 days post-infection (dpi) with recovery by 14 dpi [9-11].

Line 56—No comma after, “with”.

Line 62—DPI should be lower case.

Lines 64-67—Unclear whether this is one or two sentences.  Further, the content of these 1-2 sentences does not make sense.  Clarify.

Line 76—Should the word, “study,” be included, i.e., “…in our high dose study….”?

Lines 82-85—This statement is speculative.  Increased RBC count and hematocrit can suggest dehydration, but does not necessarily suggest decreased oxygen levels.  It’s also not clear what the authors are referring to with respect to, “compensation from hemorrhage.”  Unfortunately, the authors did not perform skin tenting on animals and/or determine total protein levels on serum obtained at time of blood draws.  If total protein is also elevated and/or animals exhibit skin tenting one can rule in dehydration as a cause of elevated RBC and hematocrit.  That would have been a very simple means of enabling clarification of this finding.  It is a leap to conclude that elevated RBC results in decreased oxygen levels—just the opposite can occur.  Without direct oxygen measurement, the authors cannot and should not be making such statements.  It’s also not clear what the authors are referring to with respect to, “compensation from hemorrhage.”  Are the authors suggesting an increase in RBC production?  If so, there would be evidence of that on the blood smear.  Since it does not appear that the authors performed direct visualization of a blood smear (again, something that could have been relatively easily and inexpensively performed and is typical of a complete clinical pathology evaluation) they would have been able to see indications of a regenerative response (i.e. RBC generation).

Lines 86-87—Write out and define IHC and NP.

Lines 89-92—Sentence makes no sense.  Clarify.

Line 102—“Comprised” not “comprising.”

Line 115-Rewrite sentence.  The word, “everyone’s,” is not appropriate for use in referring to the hamsters.

Line 116—The authors indicate that clinical signs were monitored.  What were these clinical signs, how often were they monitored, was a score sheet of some sort used to minimize subjectivity of assessment?  Specify and provide details.

Overall the materials and methods section lacks sufficient level of detail.  This section must be written in such a manner that the reader if they so choose,  can replicate the studies exactly as performed with the same reagents, equipment, timing, volumes, etc.  The authors need to go through and ensure such details are included.

Lines 120-127—Why were females 8-9 weeks and males 6-7 weeks old used?—why were they not age-matched?  What about control hamsters?  There is no mention of control animals in this section and yet Figure 1 indicates PBS was administered to controls.  Need to include procedures on all animals.

Line 123—What is unit of meausure for, “100   l.”

Line 138—What is “mid”?  Authors include in parentheses monocytes, eosinophils and basophils, but that doesn’t seem to relate.  Define and clarify.

Lines 146-153—It’s not clear where the citrate solution is used with respect to staining of the virus.  Sentence within lines 147-148 does not make sense.  Define NP.  What was the antibody conjugate used, i.e. label?  May need to rewrite this section to clarify procedures used for virus staining.  Include incubation times and temperatures.  Again, one should be able to follow the text and be able to repeat these studies exactly as performed.

Line 158—Be consistent with use of the abbreviation of days post-infection, i.e. dpi.  This is a problem throughout the entire manuscript.  Either use it or don’t use it.  If using it, then it should be lower or upper case (choose one) and should be used either before or after the day number (again choose one format).

Line 174—What is, “infection media”?

Line 182—How is viral load being enumerated?—plaque assay?—if so, indicate here that plaque assay not viral load is being analyzed.  Similarly, what is being analyzed with respect to CBC?—specify.

Figure 1—Resolution of figure is terrible.  Very difficult to read.  X axis lines for Panels A and C appear to be different weight than those in Panels B and D—should be consistent.  How was PFU/lung determined?—i.e. are there any adjustments to account for quantity of tissue evaluated or total lung weight?  Might be better to express as PFU/g of lung tissue. 

Figure 1 legend—Authors indicate that data within Panels C and D is additional data from other hamster studies not reported herein and combined with data in panels A and B.  This is a huge problem.  Nowhere in the methods is the collection and/or use of such additional data described and further no mention is made as to how such data is “combined.”  It is not appropriate to be including data from studies outside the scope of those included within the current manuscript.  If such data is to referred to it should not be included within a figure, but merely mentioned as a comparison within the text.

Line 215—3 or at most 4 digits following the decimal are necessary for expressing p-values.

Figure 2—X-axis labels in all panels should be consistent in terms of use of singular/plural day/days, use of hyphen and capitalization.

Lines 230-231—This sentence is somewhat misleading.  Perhaps revise to read, “Microscopic lesions within the lungs of each hamster were examined and scored at each time point Table S1).  Scores were combined to generate a heatmap (Figure 3).”

Figure 3—What is scale for x-axis?

Lines 267 & 270—Should be, “epithelia,” not “epithelial.”

Line 289—Not clear why Panel C is referenced here within note about Panel F. 

Line 316—Move, “J=20 um” to end, i.e. after G, H and I so in alphabetical order.  Change colon to semi-colon.  Check spacing between equal signs—should be consistent—either one space on each side of equal sign or no spaces—check all Figure legends.

Figure 9—Capitalize, “vein,” so consistent with, “Artery.”

Lines 406 & 445—Units?

Line 440—Unsure why the term “closed,” is used here.  Clarify.

Line 443-444—Inclusion of abbreviations is good, but need to also include, “M.T.”

Line 449—“COVID-19,” should be replaced with “SARS-CoV-2”. 

Line 450—It is unclear why the authors use the term, “preclinical endpoints.”  The hamster model is being developed to model the clinical (not preclinical) course of COVID-19 in humans.  Further, the use of the term, “endpoints,” should be avoided as this is a term used in in vivo animal modeling for a point (i.e. clinical or laboratory parameter) at which an animal should be terminated.

Lines 451-454—See above regarding Lines 82-85.

Line 478—What is NETosis???

Line 492—Again, reference to inclusion of additional samples.  See above regarding Figure 1 legend.  Not appropriate!!

Line 503—Why is reference 29 included within this sentence?  What is being referenced?—doesn’t seem as though a reference to this statement is needed.

Lines 522 & 524—Credit is assigned to CBJ for funding acquisition and yet it is indicated that no external funding was utilized.  This study clearly required funding.  The funding might not have been from an external source, but should be indicated—private donor, personal funds, etc.  If authors choose to not include funding source, then eliminate CBJ’s role in funding acquisition.

Line 532—Write out what RBL is, i.e., “We thank the animal care staff at the University of Tennessee Health Science Center, Regional Biocontainment Laboratory for their support of these studies.”

Author Response

The studies presented within this manuscript are important for furthering SARS-CoV-2 research.  In addition, the inclusion of data from other studies is very concerning.  Much more detail needs to be provided regarding the methodology. 

Response: We thank the reviewer for noting where additional detail may be provided and we have added this additional information to the manuscript. We have also added information about the banked paraffin-embedded samples.

The following is a list of representative issues that must be addressed before a full evaluation of the manuscript on the basis of its scientific merit, can be made--note that this is likely not a complete list and that additional issues may be present and require attention. 

Response: We thank the reviewer for the list of issues and we have reviewed and revised the manuscript accordingly. We have also reviewed the manuscript and corrected any additional issues within the manuscript.

However, the manuscript suffers from a lack of careful editing.  Further, the authors include some speculative statements that require clarification. The authors are encouraged to have one or two scientists and/or scientific technical writers not directly connected with the work, perform a thorough evaluation.

Response: We thank the reviewer for the editorial corrections. The corresponding author apologizes for not providing a thorough editorial review prior to submission and greatly appreciates the effort of reviewer 2.

Lines 39-40—Insert comma and delete, “and” so that portion of sentence reads:  “…host response, pathology and evaluation…”

Response: Thank you for the editorial suggestion. The correction has been made.

Line 47—either remove the word, “would” or change “binds” from plural to singular so portion of sentence reads:  “…RBD effectively binds….” or “…RBD would effectively bind….”

Response: Thank you for the editorial suggestion. The correction has been made to bind.

Lines 48-52—Break into two sentences.  “While the COVID-19 hamster model has provided tremendous value in the assessment of vaccines and therapeutics, standardized models remains a gap.”  [Note also that the word “remains” should be singular not plural.]  Then eliminate, “as” and start next sentence:  “Published studies have….”

Response: Thank you for the editorial suggestion. The correction has been made to remain and the sentence broken into two.

Line 55—Wording is poor and need to correctly write out and then define dpi.  Revise portion of sentence to:  “…beginning 1-2 days post-infection (dpi) with recovery by 14 dpi [9-11].

Response: Thank you for the editorial suggestion. The correction has been made.

Line 56—No comma after, “with”.

Response: Thank you for the editorial suggestion. The correction has been made.

Line 62—DPI should be lower case.

Response: Thank you for the editorial suggestion. The correction has been made.

Lines 64-67—Unclear whether this is one or two sentences.  Further, the content of these 1-2 sentences does not make sense.  Clarify. The sentences have been rewritten.

Response: Thank you for the editorial suggestion. The second sentence has been rewritten for clarity as follows.

Of course, as underscored by Gruber et al. the current knowledge regarding histopathology is from fatal human cases, and thus, translation to the COVID-19 hamster is difficult as lethality has not been reported.

Line 76—Should the word, “study,” be included, i.e., “…in our high dose study….”?

Response: Thank you for the editorial suggestion. The sentence has been rewritten.

Lines 82-85—This statement is speculative.  Increased RBC count and hematocrit can suggest dehydration, but does not necessarily suggest decreased oxygen levels.  It’s also not clear what the authors are referring to with respect to, “compensation from hemorrhage.”  Unfortunately, the authors did not perform skin tenting on animals and/or determine total protein levels on serum obtained at time of blood draws.  If total protein is also elevated and/or animals exhibit skin tenting one can rule in dehydration as a cause of elevated RBC and hematocrit.  That would have been a very simple means of enabling clarification of this finding.  It is a leap to conclude that elevated RBC results in decreased oxygen levels—just the opposite can occur.  Without direct oxygen measurement, the authors cannot and should not be making such statements.  It’s also not clear what the authors are referring to with respect to, “compensation from hemorrhage.”  Are the authors suggesting an increase in RBC production?  If so, there would be evidence of that on the blood smear.  Since it does not appear that the authors performed direct visualization of a blood smear (again, something that could have been relatively easily and inexpensively performed and is typical of a complete clinical pathology evaluation) they would have been able to see indications of a regenerative response (i.e. RBC generation).

Response: We have removed this speculation as requested in the introduction and discussion (requested below). In future studies we will evaluate total protein and use a pulse oximeter to determine oxygen saturation.

Lines 86-87—Write out and define IHC and NP.

Response: Thank you for the editorial suggestion. The correction has been made.

Lines 89-92—Sentence makes no sense.  Clarify.

Response: We have done our best to clarify.

Line 102—“Comprised” not “comprising.”

Response: Thank you for the editorial suggestion. The correction has been made.

Line 115-Rewrite sentence.  The word, “everyone’s,” is not appropriate for use in referring to the hamsters.

Response: Thank you. We have clarified the sentence as follows: Because the weights of hamsters vary, the volume of K/X injected was calculated based on the weight of the hamster.

Line 116—The authors indicate that clinical signs were monitored.  What were these clinical signs, how often were they monitored, was a score sheet of some sort used to minimize subjectivity of assessment?  Specify and provide details.

Response: This section was expanded to include the scoring system.

Overall the materials and methods section lacks sufficient level of detail.  This section must be written in such a manner that the reader if they so choose,  can replicate the studies exactly as performed with the same reagents, equipment, timing, volumes, etc.  The authors need to go through and ensure such details are included.

Response: Thank you. We have added additional details such that the experiments can be repeated precisely.

Lines 120-127—Why were females 8-9 weeks and males 6-7 weeks old used?—why were they not age-matched?  What about control hamsters?  There is no mention of control animals in this section and yet Figure 1 indicates PBS was administered to controls.  Need to include procedures on all animals.

Response: The ages differed based on availability of males and females by the vendor but were within the range of 7-9 weeks of age used in our studies.

Line 123—What is unit of meausure for, “100   l.”

Response: Thank you, the correction has been made.

Line 138—What is “mid”?  Authors include in parentheses monocytes, eosinophils and basophils, but that doesn’t seem to relate.  Define and clarify.

Response: The analyzer groups monocytes, eosinophils, and basophils together into MID designation.  This is not an abbreviation but a category that groups these cells together in a 3-part WBC differential – it’s just a function of the analyzer programming and how it sorts and counts the various WBCs.  This is also consistent with other hematology analyzers that perform 3-part vs. 5-part WBC differentials.

Lines 146-153—It’s not clear where the citrate solution is used with respect to staining of the virus.  Sentence within lines 147-148 does not make sense.  Define NP.  What was the antibody conjugate used, i.e. label?  May need to rewrite this section to clarify procedures used for virus staining.  Include incubation times and temperatures.  Again, one should be able to follow the text and be able to repeat these studies exactly as performed.

Response: The section was revised. We hope we have clarified our method.

Line 158—Be consistent with use of the abbreviation of days post-infection, i.e. dpi.  This is a problem throughout the entire manuscript.  Either use it or don’t use it.  If using it, then it should be lower or upper case (choose one) and should be used either before or after the day number (again choose one format).

Response: Thank you, the manuscript has been revised such that only dpi is used after it is defined initially in the introduction.

Line 174—What is, “infection media”?

Response: Media used to conduct the infection. This is now clarified in the text.

Line 182—How is viral load being enumerated?—plaque assay?—if so, indicate here that plaque assay not viral load is being analyzed.  Similarly, what is being analyzed with respect to CBC?—specify.

Response: Both areas of the Methods section were described further to define these terms.

Figure 1—Resolution of figure is terrible.  Very difficult to read.  X axis lines for Panels A and C appear to be different weight than those in Panels B and D—should be consistent.  How was PFU/lung determined?—i.e. are there any adjustments to account for quantity of tissue evaluated or total lung weight?  Might be better to express as PFU/g of lung tissue. 

Response: The resolution of the figure is due to the approach used by the journal in the submission of the figures within the manuscript and the total size of the document allowed. We have attached the TIFF. Figure C and D have been removed. The PFU measurement is based on the homogenization of the whole lung. We homogenize in a volume of 1 ml and use PFU/ml.

Figure 1 legend—Authors indicate that data within Panels C and D is additional data from other hamster studies not reported herein and combined with data in panels A and B.  This is a huge problem.  Nowhere in the methods is the collection and/or use of such additional data described and further no mention is made as to how such data is “combined.”  It is not appropriate to be including data from studies outside the scope of those included within the current manuscript.  If such data is to referred to it should not be included within a figure, but merely mentioned as a comparison within the text.

Response: This data has been removed and text revised as suggested. Additionally, the information and data for the banked, paraffin-embedded samples have been added to the Methods.

Line 215—3 or at most 4 digits following the decimal are necessary for expressing p-values.

Response: The numbers were corrected as suggested.

Figure 2—X-axis labels in all panels should be consistent in terms of use of singular/plural day/days, use of hyphen and capitalization.

Response: Thank you- the labels have been standardized.

Lines 230-231—This sentence is somewhat misleading.  Perhaps revise to read, “Microscopic lesions within the lungs of each hamster were examined and scored at each time point Table S1).  Scores were combined to generate a heatmap (Figure 3).”

Response: The sentence was revised as suggested.

Figure 3—What is scale for x-axis?

Response: The scale for the heatmap is shown to the right of the figure. The scale is from 0 to greater than 60%. Additional information was added to the figure legend.

Lines 267 & 270—Should be, “epithelia,” not “epithelial.”

Response: This word was changed as suggested.

Line 289—Not clear why Panel C is referenced here within note about Panel F. 

Response: To avoid confusion, we have removed this reference to the other serial section.

Line 316—Move, “J=20 um” to end, i.e. after G, H and I so in alphabetical order.  Change colon to semi-colon.  Check spacing between equal signs—should be consistent—either one space on each side of equal sign or no spaces—check all Figure legends.

Response: This image sizes were restated in alphabetical order.

Figure 9—Capitalize, “vein,” so consistent with, “Artery.”

Response: This figure was edited as suggested.

Lines 406 & 445—Units?

Response: This figure was edited as suggested; we are uncertain what happened to the units and will check the PDF on the resubmission to make sure this doesn’t happen again.

Line 440—Unsure why the term “closed,” is used here.  Clarify.

Response: Thank you, the word was modified to “close”.

Line 443-444—Inclusion of abbreviations is good, but need to also include, “M.T.”

Response: This figure was edited as suggested.

Line 449—“COVID-19,” should be replaced with “SARS-CoV-2”. 

Response: This word was edited.

Line 450—It is unclear why the authors use the term, “preclinical endpoints.”  The hamster model is being developed to model the clinical (not preclinical) course of COVID-19 in humans.  Further, the use of the term, “endpoints,” should be avoided as this is a term used in in vivo animal modeling for a point (i.e. clinical or laboratory parameter) at which an animal should be terminated.

Response: The section has been revised to ensure clarity across disciplines.

Lines 451-454—See above regarding Lines 82-85.

Response: We have omitted the suggestion.

Line 478—What is NETosis???

Response: NETosis was defined within the sentence.

Line 492—Again, reference to inclusion of additional samples.  See above regarding Figure 1 legend.  Not appropriate!!

Response: The information regarding the additional banked heart samples has been added to the Methods and Supplement and revised such that it is limited to the last section of the results in the manuscript. We apologize for our oversight of not including the details.

Line 503—Why is reference 29 included within this sentence?  What is being referenced?—doesn’t seem as though a reference to this statement is needed.

Response: This reference was removed.

Lines 522 & 524—Credit is assigned to CBJ for funding acquisition and yet it is indicated that no external funding was utilized.  This study clearly required funding.  The funding might not have been from an external source, but should be indicated—private donor, personal funds, etc.  If authors choose to not include funding source, then eliminate CBJ’s role in funding acquisition.

Response: CBJ role in funding acquisition was eliminated.

Line 532—Write out what RBL is, i.e., “We thank the animal care staff at the University of Tennessee Health Science Center, Regional Biocontainment Laboratory for their support of these studies.”

 Response: The sentence has been revised as stated.

Reviewer 3 Report

Authors presented interesting study with potential impact on clinical practise. Despite low sample size demonstrated outcome might and meaningful information to current knowledge. However, further analyses are essential.

What is potential impact on daily clinical practise with patients infected with SARS-CoV-2? Have authors planned further studies to evaluate mechanism of formation of intracardiac platelet and fibrin aggregates in the heart?

Author Response

We thank the reviewer for these questions and respond to each one below.

What is potential impact on daily clinical practise with patients infected with SARS-CoV-2?

Response: Our findings in the hamster model (with mild / moderate infection) are reflective of the pathology observed COVID-19 infection human subjects in the acute stages. However, patients with mild/moderate infection are not severely ill, thereby limiting access to the availability of specimens for histopathological analysis. We observed in our model early evidence of cardiopulmonary failure, and early thrombus formation. Further work to elucidate the beneficial effects of early institution of anti-platelet, anti-coagulant, fibrinolytic and / or thrombolytic therapy to ameliorate the prothrombotic and procoagulant state would be of significant benefit and lead to further evaluation in afflicted human subjects and has significant translational potential.

Have authors planned further studies to evaluate mechanism of formation of intracardiac platelet and fibrin aggregates in the heart?

Response: Further studies are being planned to map out the hyper-coaguable states in infected hamsters utilizing the platelet aggregation assays, D-dimer assays, thrombo-elastogram, and prothrombin, bleeding and clotting times.  Of specific interest, is the D-dimer, which is the degradation product of cross-linked fibrin and excessive fibrinolysis observed in COVID-19 infection is considered its source, along with endothelial cells damaged by inflammation. Elevated D-dimer levels in infected human subjects times is considered a poor prognostic marker and is associated with disease severity and in-hospital mortality.

Round 2

Reviewer 2 Report

Overall, the manuscript is much improved--thank you!

A couple minor things:

1-Lines 291, 690, 695 and possibly elsewhere, the authors refer to, "PBS-infected" animals.  The animals given PBS are not infected, they are control animals that have been sham-inoculated.  This should be clarified and more correctly stated.

2-Axis labels, i.e. the x-axis=Days and y-axis=Percent, should be designated on Figure 3, within the actual figure--it is not sufficient to simply indicate this within the legend.

Author Response

Overall, the manuscript is much improved--thank you!

Thank you!

A couple minor things:

1-Lines 291, 690, 695 and possibly elsewhere, the authors refer to, "PBS-infected" animals.  The animals given PBS are not infected, they are control animals that have been sham-inoculated.  This should be clarified and more correctly stated.

Response: agree. This has been corrected as suggested.

2-Axis labels, i.e. the x-axis=Days and y-axis=Percent, should be designated on Figure 3, within the actual figure--it is not sufficient to simply indicate this within the legend

Response: This has been corrected as suggested.